# Automatic precursor recognition and real-time forecasting of sudden explosive volcanic eruptions at Whakaari, New Zealand

D. E. Dempsey [1 ✉], S. J. Cronin [1], S. Mei[1] & A. W. Kempa-Liehr [1]

Sudden steam-driven eruptions strike without warning and are a leading cause of fatalities at touristic volcanoes. Recent deaths following the 2019 Whakaari eruption in New Zealand expose a need for accurate, short-term forecasting. However, current volcano alert systems are heuristic and too slowly updated with human input. Here, we show that a structured machine learning approach can detect eruption precursors in real-time seismic data streamed from Whakaari. We identify four-hour energy bursts that occur hours to days before most eruptions and suggest these indicate charging of the vent hydrothermal system by hot magmatic fluids. We developed a model to issue short-term alerts of elevated eruption likelihood and show that, under cross-validation testing, it could provide advanced warning of an unseen eruption in four out of five instances, including at least four hours warning for the 2019 eruption. This makes a strong case to adopt real-time forecasting models at active volcanoes.

[1] University of Auckland, Private Bag 92019, Auckland 1142, New Zealand. ✉email: d.dempsey@auckland.ac.nz

In the last decade, more than 1000 people were killed in volcanic eruptions[1–4] and consequent hazards, e.g., tsunami[5]. On 9 December 2019 at 2:11 p.m. (1:11 a.m. UTC), Whakaari volcano (also known as White Island) offshore of New Zealand's North Island erupted, killing or fatally injuring 21 of the 47 tourists and guides on the island. Twenty-four days earlier, Whakaari was assessed at volcano alert level (VAL) 2, on a scale of 5. This level is New Zealand's highest classification for a volcano not in eruption[6] denoting moderate to heightened volcanic unrest. Although it classifies the current state of the volcano, the VAL does not provide an eruption forecast. Expert elicitation[7,8] was used to estimate an eruption likelihood of 8–14% for the 4-week period beginning 2 December 2019, 1 week prior to the eruption[9], which represents the 50th–84th percentile range of estimates of the expert panel.

All VAL systems are inherently heuristic because, although they rely on quantitative input data, synthesis of observations to distill a threat level requires a consensus of experts, sometimes evaluating data against predetermined geophysical thresholds[10,11]. Hence the VAL number is descriptive, capturing an adjudged state (e.g., unrest, erupting) or imminent and past hazards[1,6,12]. This differs from a forecast, which is an issued probability that an eruption will occur in some future interval[13–15]. Forecasting is difficult because volcanoes are complex, individualistic systems whose eruptive character varies over time. Further, published probabilities are subject to intense post-event scrutiny, with missed eruptions or excessive false alarms eroding public trust[16].

In slow-building eruptive crises that escalate in oft-seen scenarios, VAL systems have proven useful[1,12]. However, for small-scale eruptions from volcanoes in a semi-continuous degassing or hydrothermally active state, the bottleneck of human consensus is too slow to issue warnings. Outside of a crisis, the organizational burden of frequent reassessment means that VAL classifications persist for extended periods. Furthermore, use of aggregated subjective opinion makes it difficult to establish chains of reasoning linking input data and output recommendations; e.g., phreatic eruptions at Whakaari produced immediate post-event alert level changes from 1 to 2, from 1 to 3, and from 2 to 4, with similar geophysical or observational data in each case (Supplementary Table 1). We can address these issues by complementing VAL with automated systems that are rapid, high-resolution, and transparent.

Research into volcanic eruption precursors has largely focused on tremor[17], a continuous, long-period seismic signal in the 1–15 Hz frequency band. Anomalous tremor signals have been suggested to originate from fluid–rock interactions in magma chambers and overlying hydrothermal systems, including oscillation in cracks and conduits[18,19] or accumulation of gas[20]. Tremor data are typically processed in specific frequency bands, for example, real-time seismic amplitude measurement (RSAM) or displacement seismic amplitude ratio (DSAR), which can then be used in short-term forecasting[20,21]. In particular, failure forecast modeling (FFM) of tremor has shown promise at Whakaari[21] and was successfully applied in foresight to the 1998 eruption at Colima (Mexico)[13].

Machine learning methods are an emerging tool across the geosciences[22,23] as scientists adapt to large, complex, multidisciplinary data streams and physical systems. For example, unsupervised pattern matching has been used to identify eruption precursors at Mt Etna[24] although attempts to classify eruption onset have been unsuccessful[25,26]. In seismology, time series feature engineering has revealed links between seismic tremor signals and slip on subduction zones[27]. Here, we detail a machine learning pipeline that combines precursor extraction from volcano tremor data, classification modeling of eruption imminence,

and a decision model that issues eruption alerts. Although only a prototype, our model learned to correctly identify eruption precursors and used these to issue an advanced warning for eruptions it had not previously seen. Under pseudoprospective testing conditions, it did this for four out of five recent eruptive episodes at Whakaari, including the fatal 2019 eruption.

## Results

**Machine learning identification and modeling of precursors.** Continuous seismic data are available at Whakaari in near real-time from a broadband station (WIZ, Fig. 1a) located about 1-km southeast of the main vents. We processed 9 years of data, from 1 January 2011 to 1 January 2020 (Fig. 1b), to obtain four time series that capture different parts of the tremor signal (RSAM, DSAR, medium and high frequency bands—MF and HF—sampled every 10 min; see "Methods"). The dataset spans five major impulsive eruptive episodes: (1) an eruption on 4 August 2012 at 16:52 (all times in UTC) that ejected ash and blocks[21]; (2) a steam and mud eruption on 19 August 2013 at 22:23; (3) three eruptions beginning with energetic steam venting 3 October 2013 at 12:35, followed by a minor mud-steam eruption October 8 and culminating in a moderate explosive eruption on 11 October (Supplementary Table 1); (4) six eruptions over 35 min beginning on 27 April 2016 at 09:37[28]; and (5) the recent fatal eruption on 12 December 2019 at 01:11. We excluded episodes of minor eruptive activity from our analysis, e.g., geysering, lava dome growth[21], and passive ash emissions, because we are principally focused on classifying hazardous impulsive/explosive eruptions with the highest threat to visitors on the island.

We identified eruption precursors in the Whakaari tremor data using systematic time series feature engineering techniques developed for anomaly detection in industrial steel casting[29]. First, we sliced the tremor data into 48-h windows that overlapped by 36 h. Second, for each window, we computed 706 features of the 48-h time series, e.g., mean and standard deviation, slope and standard error of a linear regressor, and Fourier coefficients[30]. Third, we assigned each window a binary label indicating whether an eruption did (1) or did not (0) occur in the 48 h following the window (Fig. 1c).

Next, we searched for features whose values clearly indicated an association with imminent eruption: these are the precursors. This was done by applying a statistical test to the distribution of each feature (over all windows) and identifying those features whose values tagged with a 1 were significantly different to those tagged 0 (see "Methods" for details). Notable features included the Fourier coefficient for 40-min periodicity in RSAM, which had a large value before four out of five eruptive periods (Fig. 1d), as did RSAM maximum value and 4-h periodicity (Fig. 1e, f). In contrast, the gradient of a linear regressor fitted to 1/RSAM (Fig. 1g) was less remarkable. This parameter is used to extrapolate eruption onset in FFM[21] but needs to be complemented by other indicators of eruption imminence to be useful. We ranked features by $p$ value and computed a Benjamini–Yekutieli correction[31] to check for excessive false discovery (Supplementary Fig. 1).

Finally, we formulated the precursor-eruption relationship as a classification problem. We trained a decision tree to classify significant features by value as eruption-imminent or no-eruption. Decision trees have simple branching structures that evolve to optimally partition 0 and 1 labels according to their corresponding feature values. We used these models because they can accommodate dissimilar looking eruptions (from a precursor/feature sense) on separate branches. Our dataset contained many more 0's than 1's, hence, before training the model, we balanced it by randomly discarding some no-eruption windows. This discard step is arbitrary and so we repeated it 100 times to generate and

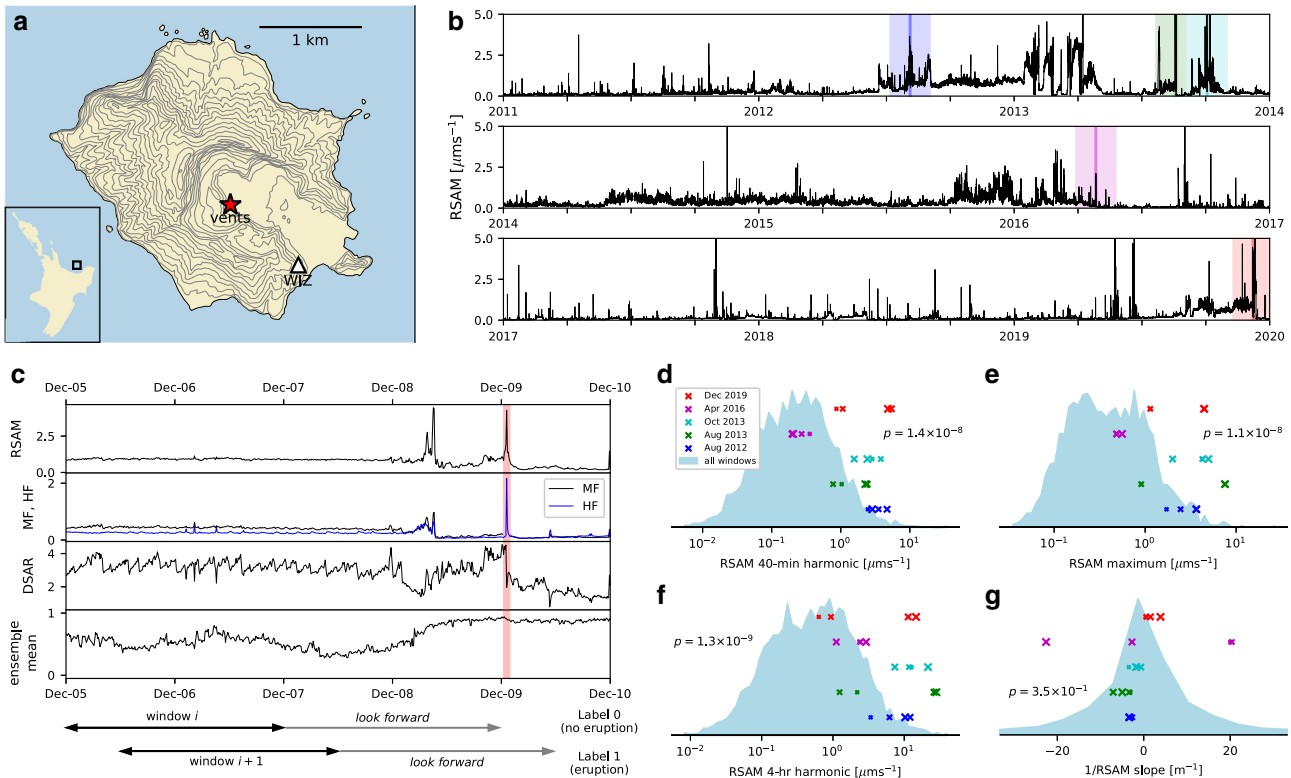

**Fig. 1 Whakaari tremor data and features. a** Location of vents and seismic station at Whakaari volcano. **b** RSAM (Real-time Seismic Amplitude Measurement) tremor signal over the 9-year study period, with five eruptive periods indicated by darkened colored lines. The wide-shaded bar either side of each eruption demarcates the testing interval during cross-validation. **c** Tremor time series (RSAM, MF, HF, DSAR) in the 4 days preceding the December 2019 eruption, indicated by the vertical red bar (UTC time). The relative positions of two adjacent windows (black arrows at bottom of **c**) and their associated look-forward periods are indicated below. The eruption falls outside the look-forward period (gray arrow) of window $i$, but inside the look-forward of $i + 1$, and are thus labeled 0 and 1, respectively. The ensemble mean of an eruption forecast model that accepts the windowed tremor data is given in the bottom frame of (**c**). **d–g** Frequency distribution of exemplary feature values, with values prior to the five eruptions (colored markers, corresponding to colored lines in (**b**) explicitly plotted. **d–f** show statistically significant features for eruption forecasting. Larger markers denote a window nearer to the eruption. Mann–Whitney $U$ $p$ values are quoted and indicate feature significance.

ensemble of 100 different decision trees. This bagging approach implements a random forest specialized on eruptive windows, and the ensemble mean prediction is used to issue alerts.

**Forecasting Whakaari eruptions.** When operating, our forecaster computes time series features from the previous 48 hours of tremor data and passes these to the random forest. This yields 100 predictions of whether an eruption will occur (1) or not (0) in the next 48 hours. Our forecast model updates every ten minutes. The forecast is passed to a decision model that determines whether an alert should be issued. Here, we issue an alert if the ensemble mean exceeds a threshold, and then rescind it when it stays below the threshold for 48 hours. If no eruption occurs during the alert, then it is a false positive (FP). If an eruption occurs without an alert issued, it is a false negative. These errors trade-off against each other: as the alert threshold is reduced, we are less likely to miss future eruptions but at a cost of more FPs (Fig. 2a). Not all alerts will contain eruptions, so the true positive (TP) rate is an estimate for the probability that an eruption will occur during an alert. This is an important figure for public communication.

We checked our model's predictive capability through cross-validation (Supplementary Fig. 2), a pseudoprospective test that mimics aspects of real forecasting (but is, strictly speaking, hindcasting). Data from 1 month either side of the December 2019 eruption were reserved in a testing set, and the remaining data were used to select features and train the random forest.

The test data were then input to the forecaster that returned an ensemble mean prediction over the 2-month period. Only prediction of the eruption in the test data is meaningful. We computed the threshold that would have been required for an alert to have been issued in advance of the eruption. Assuming this threshold had been universally adopted from 2011 to 2020 (Fig. 2b), we computed the number of issued alerts and the probability for an eruption to occur during an alert (Fig. 2a). We then repeated these steps for the other eruptive periods in the dataset, which yielded five separately trained models and forecasting tests (Fig. 2c).

For this pseudoprospective test, we found that a threshold of 0.8 (80 out of 100 models predicting an eruption) provides advanced warning for five out of the last seven Whakaari eruptions. This includes a 17-h warning of the fatal 2019 eruption, with a peak ensemble mean of 0.94 occurring 4-h prior to that event. For the 4 August 2012 eruption, an alert was issued 16 h prior, several days after an alert raised on 1 August. For the 19 August 2013 eruption, an alert was issued 75 h prior. For the October 2013 eruptive period, 7-h advanced warning was issued for the first (3 October) eruption, the 8 October eruption was missed, and a 70-h warning was issued for the largest eruption on 11 October. The middle eruption in this sequence had anomalously high RSAM activity in the 24 h prior (Supplementary Fig. 3A) that would likely have concerned a human operator. However, the signature differed from other eruption precursors

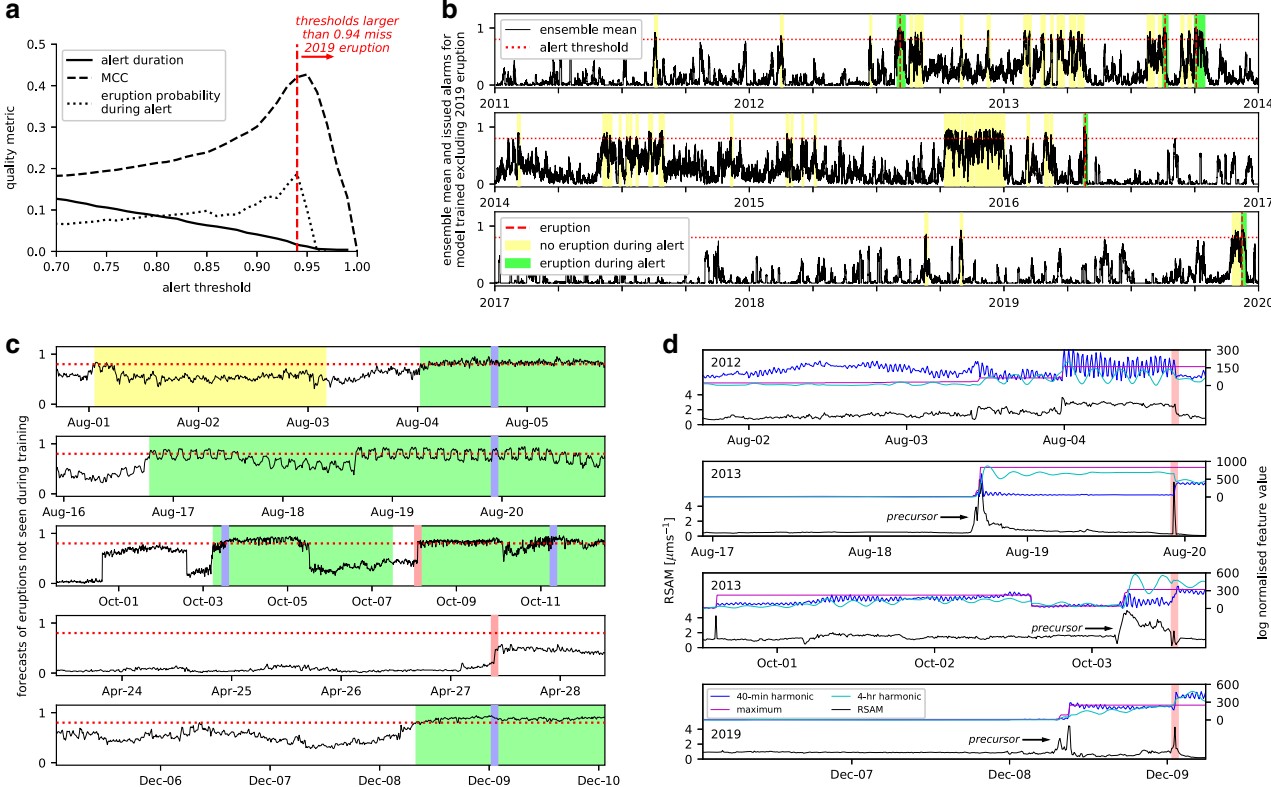

**Fig. 2 Whakaari eruption forecast model. a** Quality metrics as a function of alert threshold for a model trained excluding the December 2019 eruption: MCC = Matthews correlation coefficient, a balanced quality metric similar to $r^2$ (see "Methods"); eruption probability during alert = proportion of all raised alerts that contain an eruption; alert duration = fraction of analysis period during which the forecast model is in-alert. The red dashed line indicates an alert threshold above which the December 2019 eruption would have been missed. **b** Performance of the same model over the analysis period for a threshold of 0.8 (red dotted line). Ensemble mean (black), eruptions (vertical red dashed lines), and alerts with (green) and without eruptions (yellow) are shown. **c** Performance of forecast models under cross-validation, anticipating four out of five eruptive periods (five out of seven eruptions) when that eruptive period is excluded from training. As in (**b**), the alert threshold is 0.8. Missed eruptions are indicated in red, the remainder in blue. **d** RSAM signal (black) in the three days prior to the 2012, 2013, and 2019 eruptions, alongside the three feature values from Fig. 1d–f (blue, magenta, cyan). To aid comparison, feature values have been normalized in log space to zero mean and unit standard deviation. Precursor signals identified by arrows are referred to in the text.

enough that the forecast model did not issue an alert. The 27 April 2016 eruption is missed by the forecaster entirely.

At a 0.8 threshold, the model issued 58 alerts that covered 8.5% of the 9-year study period. With five of the alerts containing eruptions, the eruption probability during an alert is estimated at 8.6%. A more specific model, using a 0.93 threshold, generates 26 alert periods covering 2.5% of the study period, with a corresponding 15% eruption probability during alert. However, with this formulation the 3 October 2013 eruption is missed. In the absence of perfect data and models, these are the trade-offs that policy-makers, regulators, and emergency managers must grapple with.

The leave-one-out method of performance evaluation only yields a theoretical estimate of the model's present accuracy based on all data. However, if the forecast model were in operation since the 2012 eruption, its performance would be different because it would have had access to less training data. To emulate this scenario, we trained four models using data up to 1 month prior to each of the eruptive periods two (August 2013) through five (December 2019), and then generated a forecast for the eruptive period (Supplementary Fig. 4). Both the October 2013 and December 2019 eruptions are anticipated at a 0.8 alert threshold based on knowledge of previous events. In contrast, observation of the 2012 eruption is not sufficient to pick the August 2013 eruption.

**Eruption precursors at Whakaari.** Not all eruptions have similar precursors. The missed 2016 eruption is much weaker at 40-min and 4-hr RSAM periodicity that characterizes the other eruptions (Fig. 1d–f), and exhibited low tremor before and after the event (Supplementary Fig. 3B). The VAL prior to this eruption noted minor volcanic unrest, and gas flux and surface temperatures were normal or slightly elevated, in contrast to the 2012 and 13 eruptions levels of gas flux and surface activity were rising notably (Supplementary Table 1). A recent study[32] of ejecta from the 2016 eruption found evidence of hydrothermal mineralization and a hydraulic seal in the shallow vent area, along with no magmatic ejecta. These factors indicate that the event was a pure hydrothermal explosion of trapped meteoric fluids, without any increased injection of fluids into the system from below.

The missed event emphasizes a limitation of data-driven models, which is that they only anticipate eruptions with precursors similar to those seen before. Future refinements to address this issue could include training the model using other data streams, e.g., continuous gas monitoring or shallow sensors installed near the vent. Adding precursory signals from other volcanic systems (e.g., via Bayesian approaches[33]) could also help, although this would require testing and evaluation to ensure the unique signals identified for a particular volcanic system are not diluted. Ultimately, nuanced discussion of different eruption types and pre-eruption conditions (e.g., gas and surface

observations) should always be considered to provide context and balanced when forecasting future events.

We have used the values of significant features to highlight macro elements in the tremor time series that our model regards as eruption precursors (Fig. 2d). The 2019, and both 2013 eruptions are preceded by strong RSAM bursts—paroxysms— some 4 h in duration, and overlaid by a shorter, 40-min oscillation. Such signals have been associated with degassing instabilities in volcano-hosted hydrothermal systems[18], recording convection and gas cycling that is modulated by transient sealing of gases below mineralized zones[34], muds, or sulfur pools below crater lakes[35]. The 4-h paroxysmal signal could indicate rising magma into the lower conduit, which further charges the hydrothermal fluids[36,37]. This hypothesis is consistent with extrusion of a lava dome following the 2012 eruption, high rates of $SO_2$ emission following 2013 and 2019 eruptions, and recent lava sightings after the 2019 event. The magmatic gas drives additional overpressure and failure through the hydrothermal system[38,39], with paroxysm duration dependent on the rate of gas input, and the permeability and strength of the hydrothermal reservoir and capping materials.

## Discussion

Data-driven forecasting has several advantages that complement heuristic VAL systems. First, it is rapid: on a desktop computer, it takes less than a minute to download the latest tremor data and compute a forecast. The result could be promptly uploaded to a public website or, communicated directly to VAL operators, which is valuable both prior to and during a volcanic crisis[40]. Second, they provide an objective way to estimate eruption probability during the alert period. Third, these models are dispassionate in their evaluation of data, unencumbered by social or political pressures and imperfect models of idealized eruption precursors.

In automating key decisions such as feature (physics) selection, classifier calibration, and cross-validation, we guard against model overfitting. To illustrate the costs of these choices, we developed a competing forecast model in which we hand-picked two features (RSAM maximum and 4-h harmonic, Fig. 1d, e) and issued an alert whenever they both exceeded thresholds. This two-node decision tree is optimized on the entire dataset, i.e., with no cross-validation or pseudoprospective testing (Supplementary Fig. 5). This manual model issued twelve alerts in the 9-year study period, which contained six out of the seven eruptions (April 2016 is missed). The alert duration was 55 days and the eruption probability during alert was 50%. Superficially, this performance is significantly better than the ensemble model. However, as overfitting has been maximized, its prospective performance on future data is likely to be very poor.

Automatic forecasting models have their drawbacks as well. Overfitting can also be caused by software errors or data leakage[41] that inflate accuracy and tempt an operator to set an overly optimistic alert threshold that misses eruptions. The inability to recognize unfamiliar unrest signatures could result in failure to issue a timely alert, whereas a human operator might recognize unusual activity. To integrate such models with VAL, sufficient resourcing would be needed for development, quality assurance, operation, and personnel training. This is on top of the resourcing necessary to install and maintain a telemetered seismic station at a remote location for extended periods. Extensive monitoring networks are the privilege of well-funded volcano observatories, but this system can run only with a single station to reduce costs. Finally, there is unavoidable designer bias in selecting the classifier (decision tree) and the data source (RSAM from WIZ station). Further, our choice of 48-h data windows and look-

forwards represents an optimization of the model performance on the available data (Supplementary Fig. 6). In other fields, this has led to spurious identification of precursors, e.g., accelerating moment release prior to large earthquakes[42]. Cross-validation constrains these biases, however the gold standard for forecasting is evaluation of a frozen model on future data[43].

This forecasting approach does not resolve the problem of who decides when to publicize warnings. Our forecast model issues a warning with an estimated eruption probability that is conditional upon a specified threshold. Choosing this value is a trade-off between alert duration, accuracy, and specificity, with implications for public safety, economic activities, and public trust. Deciding who is responsible for setting these thresholds is a societal debate, as is determining access to the model outputs. Ultimately, tour operators, government regulators, and the public all bear some responsibility for adjusting their actions in response to new information about the volcano state. These issues must be urgently addressed if automatic forecasting is to meet societal expectations of volcano warnings and prevent future tragedies.

## Methods

**Data collection and processing.** Raw seismic velocity data, $v$, were downloaded using the ObsPy API[44] for the WIZ broadband GeoNet station located 1-km southeast of the main Whakaari vent. A second station on the island, WSRZ, did not provide data coverage across all eruptions and was not considered. Data were downloaded for the period 1 January 2011 to 1 January 2020. The instrument response was removed and the signal bandpass filtered to isolate three frequency ranges: RSAM[45] between 2 and 5 Hz, $v_{RSAM}$; MF between 4.5 and 8 Hz, $v_{MF}$; and HF between 8 and 16 Hz, $v_{HF}$. Each filtered time series was incremented into 10-min, non-overlapping windows and the average absolute signal amplitude computed, yielding $\bar{v}_k$ where $k$ denotes the frequency band. A fourth time series, a modified DSAR[20], was computed by integrating the MF and HF signals, computing the average absolute signal over 10-min windows, and then taking the ratio of the two quantities, $\bar{u}_{DSAR} = \bar{u}_{MF}/\bar{u}_{HF}$ where $u_k = \int_k^v dt$. Three transformations of the four time series $\bar{v}_k$ and $\bar{u}_{DSAR}$, collectively $X$, were computed: inverse, $1/X$; derivative, $\dot{X} \approx (X_{i+1} - X_i)/(t_{i+1} - t_i)$; and logarithm, $\log_{10}X$. These did not improve forecasting ability and are not reported on further. There are about 85 days of missing data, or 2.5% of the analysis period. Time series gaps were imputed by linear interpolation and, because these gaps are restricted to noneruptive periods, this choice had no impact on our analysis.

The five major eruptive episodes at Whakaari considered were: (1) a phreatic eruption ejecting ash and blocks on 4 August 2012 at 16:52 (UTC), (2) a minor steam eruption 19 August 2013 at 22:23, (3) a minor phreatic eruption 3 October 2013 at 12:35 with a subsequent steam and mud eruption 8 October 02:05 and a moderate explosive eruption 11 October 07:09, all classified as a single eruptive episode, (4) a moderate phreatic eruption comprising six events in a 35-min period beginning 27 April at 09:37; and (5) the most recent 12 December 2019 at 01:11. In addition, lesser periods of unrest and minor eruptive activity include lava dome growth September to November 2012, intermittent mud and ash eruptions January to April 2013, and a minor ash emission September 2016[46] (Supplementary Table 1). As these episodes are not significant impulsive eruptions and do not pose a hazard to tourist visitors, they were excluded from the analysis.

**Time series windowing, labeling, and feature extraction.** Time series data were sliced into overlapping windows of length $T_w$ and overlapping fraction $\eta$, which yielded $N_w$ total windows for each time series $X$. From the end of each window, a look-forward interval of length $T_{lf}$ was inspected for eruptions, with the window labeled 1 if an eruption occurred and 0 otherwise. Windows were timestamped with their last data point—the end of the window and the beginning of the look-forward—which ensured temporal demarcation between the future (an eruption, or not) and the past (time series features), thereby guarding against data leakage[41].

Automatic time series feature extraction was performed using the Python package tsfresh[30], which calculated $N_{ft} = 759$ features from each window, including distribution properties (e.g., mean, standard deviation, kurtosis), measures of autocorrelation (e.g., fast Fourier transform (FFT) and power spectral density coefficients), properties of linear regressors (e.g., gradient, standard error), energy, entropy, and stochasticity. For a window with $n$ samples, FFT coefficients are computed for frequencies $[1, \ldots, n/2 - 1]/(T_w n)$, which for a 48-h window with 10 min samples correspond to $8.3 \times 10^{-4}$ to $5.8 \times 10^{-6}$ Hz. Time series features were stored in a matrix $M$ comprising $N_w$ rows and $4 \times N_{ft}$ columns, with each column charting the evolution of a feature over the analysis period, and each row characterizing the volcano state at a given time. Window labels were stored in the vector $Y$, which has length $N_w$.

The Mann–Whitney $U$ test was applied to each feature (columns in $M$), testing whether values corresponding to windows labeled 1 were likely to have been drawn

from a distribution having a different median than the distribution labeled 0. Each test yielded a $p$ value which allowed us to sort features from smallest (most likely to be significant) to largest (least likely).

**Classification modeling.** A test dataset was constructed by removing rows (windows) from feature matrix $M$ 1 month either side of an eruption (Supplementary Fig. 2B), yielding two feature matrix/label vector pairs, $[M_{train}, Y_{train}]$ and $[M_{test}, Y_{test}]$. Due to the relative infrequency of eruptions in $Y_{train}$, there was a substantial imbalance between 0 and 1 labels: for typical values $T_w = 2$ days and $\eta = 0.75$, there were 6453 entries in $Y_{train}$ of which only 16 were labeled 1. We rebalanced the dataset by randomly sampling (without replacement) 21 of the label 0 windows (0.3%) from $M_{train}$ and $Y_{train}$ and discarding the rest. Features and labels of the reduced set of $N_{ru} = 37$ windows (Supplementary Fig. 2C) are denoted $[M_{ru}, Y_{ru}]$. The undersampling ratio is the ratio of eruptive windows to noneruptive windows, in this case 0.75. As the undersampling step was arbitrary, we repeated it $N_{cl} = 100$ times to obtain an ensemble of rebalanced datasets, the $i$th denoted $\left[M_{ru}^{(i)}, Y_{ru}^{(i)}\right]$. Across all the classifiers, about 30% of the noneruptive data are used.

Hypothesis testing and feature ranking was performed on each pair $\left[M_{ru}^{(i)}, Y_{ru}^{(i)}\right]$ and the 20 most significant features were selected for classification modeling. We sought the optimal algorithmic transformation $f$ with associated hyperparameters $\theta^{(i)}$ that maximized the balanced accuracy, defined as the arithmetic mean of sensitivity and specificity. We tested seven different classifier models implemented in the scikit-learn Python library[47]: Decision Tree, k-Nearest Neighbors, Support Vector Machine, Logistic Regression, Random Forest, Naive Bayes, and Neural Network. Classifiers were trained with the best $\theta^{(i)}$ determined by a grid search (Supplementary Table 2) with Shuffle & Split (S&S) cross-validation. In S&S, a random 75% of the $N_{rs}^{(i)}$ windows are used to train the classifier, and the other 25% used to test its performance (Supplementary Fig. 2D). This was repeated five times for different train-test splits, and the best hyperparameters $\theta^{(i)}$ returned for that classifier. The decision tree consistently outperformed other classifiers in model confidence of imminent eruption while also minimizing total alert duration (Supplementary Fig. 7). Possibly this is because, like event trees[33], the hierarchical structure of decision trees can naturally accommodate different kinds of eruption precursors. Only results using decision tree classifiers are reported here.

We identified highly correlated features by inspecting scatter cross plots of the top 50 most-common features used in classification. Highly correlated features were due to different methods for calculating linear trends and we eliminated these by dropping their columns (53 features) from $M$ and retraining the classifiers. The $p$ values of features selected for classification modeling are summarized in Supplementary Fig. 1 and range between $10^{-10}$ and $10^{-2}$. When performing large numbers of statistical tests and using $p$ values to select features, the false discovery rate can be controlled by applying a Benjamini–Yekutieli correction. Supplementary Figure 1 shows that 99.3% of our features have $p$ values that are less than a false discovery threshold of 5%. Excessive false discovery of features would lead to model overfitting where classifiers are erroneously trained to replicate noise in the data. Our model anticipates five out of the seven most recent eruptions for a total alert duration of 8.5%. To achieve the same success rate by random allocation of alerts would require a total alert duration of 70%. This indicates that any potential overfitting is not debilitating to the model's performance.

**Eruption forecasting and quality metrics.** We constructed an eruption forecast at time, $t_j$, by computing time series features, $M^{(j)}$, for the new data window $[t_j - T_w, t_j]$. These features were passed as an input to each of the $N_{cl}$ trained decision trees, $f^{(i)}$ (Supplementary Fig. 2E, F). We defined the ensemble mean, $\bar{y}^{(j)} = \sum_i f\left(M^{(j)}, \theta^{(i)}\right)/N_{cl}$. We cannot interpret this as a probability of a future eruption, because the models are not individually or collectively calibrated. If the ensemble mean exceeded a threshold, $\bar{y}^{(j)} > \bar{y}_{crit}$, an alert was issued at $t_j$. An alert was only rescinded after a continuous period $T_{lf}$ in which $\bar{y}^{(j)} < \bar{y}_{crit}$, the model saw no sufficient agreement of eruption in the next $T_{lf}$. Therefore, alert periods had a minimum duration of $T_{lf}$ but could have been longer depending on the persistence of identified precursors. Although the model was trained with windows that overlap by 75%, when used for forecasting (i.e., applied to test data during cross-validation) we provided updated windows every 10 min. This is important because if a precursor occurs <12 h before an eruption, it could be missed by a forecast model updating every 12 h. Alerts were issued more regularly as forecast resolution was increased from 12 h to 10 min.

Classifiers were evaluated by comparing known, $Y$, and modeled, $f(M)$, label vectors. We converted our nonbinary ensemble mean, $\bar{y}$, to a binary label, $\bar{y} > \bar{y}_{crit}$ and compared against $Y$. We scored our eruption model by computing the Matthews correlation coefficient (MCC, as implemented in scikit-learn). MCC is a goodness-of-fit metric for imbalanced classification problems, and ranges between 1 for a perfectly accurate model and $-1$ for a perfectly inaccurate model, with a score of 0 indicating guessing. The MCC for our model varies between 0.2 and 0.4 (Fig. 2a) depending on $\bar{y}_{crit}$, indicating a weak positive to moderate positive relationship. As a metric, MCC assigns the same weight to a missed eruption as it

does to an alert that does not contain an eruption. In practice, we expect that the general public are likely to assign greater significance to missed eruptions.

We used two other performance measures to score the decision model (issued alerts, as opposed to $\bar{y} > \bar{y}_{crit}$). First, we counted a TP each time an eruption fell inside an issued alert, i.e., the eruption had been correctly anticipated by the model. We counted a FP for each alert that the model had expected an eruption to occur but did not eventuate. We then defined the eruption probability during alert as TP/ (FP + TP), which is the probability that an eruption will occur within a randomly selected alert. Second, we computed the total alert duration, which is the fraction of the 9-year study period for which Whakaari was under an issued alert.

In addition to these performance metrics, forecasts were scored on their confidence in issuing advance warning of eruptions not seen during training. We defined model confidence as the highest threshold that would have raised advanced alert of an eruption the model had not yet seen. This is computed as $\max(\bar{y}([t_e - T_{lf}, t_e]))$ for an eruption occurring at time $t_e$.

All our models were trained excluding the data 1 month either side of a test eruption, and therefore only our forecast of that test eruption is meaningful. The other eruptions in the training set were generally anticipated because that is explicitly what the forecaster was taught to do. The same is not true for noneruptive periods. Because of the random undersampling, 69% of noneruptive windows were never selected for training in any of the 100 decision trees, and only a small minority were selected to train more than one classifier. Thus, our model provides a genuine forecast during noneruptive periods across the entire dataset, and this interval can be used to quantify the eruption probability during alert.

**Sensitivity of the forecast model to design decisions.** If the forecast model is learning properly from data, then its performance should improve as the amount of data available for training increases. We trained four different forecast models using data between 1 January 2011 and 1 month after the $i$th eruptive episode where $i$ ranges between 1 and 4. Each trained forecaster was scored by its confidence anticipating the 2019 eruption as well as the total alert duration at 0.8 ensemble mean threshold (Supplementary Fig. 8). Model confidence generally increased as information about additional eruptions was incorporated. Alert duration also increased, which was unexpected. This is because the forecasters trained with fewer eruptions were less equipped to recognize and raise alerts during noneruptive episodes of volcanic unrest, e.g., February to April 2013, June to August 2014, and October to December 2015.

In addition, three variations of the fourth model above were developed, each reflecting a different protocol for model retraining: (i) model was retrained at the beginning of each year, i.e., using data from January 2011 up to January 2019, (ii) at the beginning of each month, i.e., up to December 2019, and (iii) at the end of each look-forward period, i.e., up to 7 December 2019. The incorporation of additional noneruptive data had a weak effect on model confidence and alert duration (Supplementary Fig. 8).

Selecting a window length, $T_w$, is a trade-off between different model performance metrics (Supplementary Fig. 6A). We developed a set of forecast models in which $T_{lf}$ was fixed to 48 h, and $T_w$ was varied between 12 h and 5 days. Each model was trained four times, excluding in turn each of the 2012, two 2013, and 2019 eruptive episodes, and then averaging forecast confidence anticipating the excluded eruption. Average model confidence decreased for longer windows because these diluted the precursor signal with non-precursory information. Whereas, for very short window lengths the forecaster became overconfident, issuing many more alerts. We adjudged $T_w = 48$ h as a good trade-off between these two effects.

Similarly, we fixed $T_w$ to 48 h and varied $T_{lf}$ between 12 h and 5 days, retrained the models and computed average confidence forecasting the same four unseen eruptions (Supplementary Fig. 6B). Model confidence was highest for a 48-h look-forward and was eroded for comparatively short-sighted and long-sighted models, although instability at shorter look-forward periods is noted. Increasing the look-forward automatically increased the length of issued alerts, and hence increased total alert duration.

The performance of the model forecast is similar under variation of the undersampling ratio (Supplementary Fig. 9). Reducing this value includes more noneruptive data in the training subsets, which decreases model alert duration. This is similar to the trend identified in Supplementary Fig. 8B, i.e., a reduced specialization on eruptive data means that fewer alerts are identified during noneruptive episodes of volcanic unrest.

The performance of our model forecast model is not adversely sensitive to arbitrary seeding of the random number generator (RNG) that controls undersampling, and S&S cross-validation. For models retrained using five consecutive integer seeds of the RNG, model confidence, alert duration, and ensemble mean are all similar (Supplementary Fig. 10).

## Data availability
The processed tremor data to replicate the results of this study are available at https://github.com/ddempsey/whakaari.

## Code availability
The codes required to replicate the results of this study are freely available at https://github.com/ddempsey/whakaari, released under the MIT license.

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

## Acknowledgements

Tremor data were supplied by the GeoNet seismic monitoring system. Whakaari map data provided by Land Information New Zealand. The authors thank M Bebbington for reviewing an early draft of the manuscript. S.J.C acknowledges support of MBIE Smart Ideas project "Stable geothermal power generation with reduced hydrothermal hazard" UOAX1807.

## Author contributions

D.E.D. and A.W.K.-L. designed the eruption forecasting study. S.M. implemented an early prototype of the forecaster. D.E.D. implemented the final forecast model. S.J.C. compiled Whakaari VAL data and GeoNet bulletins. All authors contributed to manuscript development.

## Competing interests

The authors declare no competing interests.

**Additional information**

