## [Peer Review File · Nature Communications]

REVIEWER COMMENTS

Reviewer #1 (Remarks to the Author):

Review of the paper

RECOGNIZING THE WARNING SIGNS OF VOLCANIC CATASTROPHE

D E Dempsey, S J Cronin, S Mei, A W Kempa-Liehr

This paper addresses one important scientific and social issue, such as the forecast of "blue-sky" eruptions; so far, these eruptions are barely predictable causing many serious problems, mainly for tourists.

The paper is very well written and clear; it was a nice reading to me. The technical content of the paper is of high quality and it introduces a novel promising tool to improve our blue-sky eruption forecasting skill.

At the same time, as admitted by the same authors in this manuscript, the evaluation of the forecasting skill of the proposed method is still very speculative, being based on hindcasting, which often leads to significant overfitting of the data and too optimistic conclusions.

So, in general, it is important to make this distinction crystal clear for the readers, also in the abstract; to me this would not downplay the possible impact of the paper. As a matter of fact, the paper is more methodological and the testing phase is still very preliminar. I understand that it takes time to evaluate a model in a prospective way (seismologists have exactly the same problem with large earthquakes), but it is the gold rule to evaluate the real predictive skill, and I do not know any shortcut.

Below I list the specific comments on this paper

- The title probably is too emphatic, in the sense that the paper addresses one specific kind of eruptions (blue-sky) which are very interesting from the scientific point of view, but usually impact a limited number of tourists. It goes without saying that this is not meant to under-evaluate the life of each single person, but probably the word "catastrophe" may be used for different types of eruptions.

- As corrected mentioned by the authors, the VALS in many volcanoes (and certainly for Whakaari volcano) are **not** a forecasting scheme, even though it is often interpreted as such. Actually, in the volcanological literature it is possible to find critics to VALS (and more in general on how volcanologists manage phases of unrest), because under a VALS approach volcanologists tend to act as decision makers without having proper competences to decide, for example, "when" it is the right moment to call for an evacuation and how long it should last; I think we, as scientists/volcanologists, have to pay attention to these aspects if we do not want to face harsh critics and possible legal prosecutions after any kind of subsequent formal enquiry (it would not be the first time in natural hazard...). So I think we, as scientists/volcanologists, should be very careful in suggesting any kind of "consensus" thresholds in the forecasts (see LINES 177-178), because they do not have any scientific meaning but they can be justified only in a decision-making world, e.g. through some sort of cost/benefit analysis for risk mitigation decisions. Of course, the method proposed by the authors can be integrated in a VAL, but I would make very clear the distinction between the forecasting tool proposed by the authors and the integration into a VALS. This mapping requires different competences. The authors seem to recognize this crucial point at lines 139-140, but at the end of the paper (LINES 171-180), it seems that they advocate some thresholds chosen through a consensus (of volcanologists? who is the "designer" that has to chose this?).

- LINES 104-107. The authors acknowledge the problem of handling sequences in which we have many more "no eruption" than "eruption" windows. There is not a unique way to handle this problem (for instance, one possible strategy is to weigh in a different way the correct predictions and the false alarm rates for each category). The authors decide to take 100 subsets in which the "no eruptions" windows are randomly discarded to make the numerosity of each class more comparable. It is not clear to me how this step is carried out. What is the proportion of "no eruption" windows that are discarded for each subset?

If we want to have similar number of "eruption" and "no eruption" windows I guess that the percentage of discarded windows is very large (even though the authors say they discard "some no eruption windows"). In any case I am wondering if different choices of the percentage of discarded windows may affect the results.

- LINES 145-149. The authors say that data-driven models can only anticipate eruptions whose precursors have been seen before. The authors forget that, nowadays, many eruption forecasting schemes are not full data-driven models. For example, many implementations of the Bayesian Event Tree model take already into account the possibility to have different precursors from eruption to eruption, and precursors that have not been observed before.

- LINES 163-170. The authors emphasize the importance to have "real-time" tools to forecast the evolution of the volcanic system, compared to the slow process of getting forecasts from experts' judgment elicitation. I do agree with that. But probably it should be said that this point has been raised several times in the past, also for New Zealand volcanoes (Lindsay et al., 2010).

- LINES 280-283. The authors use the Mann-Whitney U test to check if two distributions are different. I think that the Mann-Whitney U test is tailored to check if the "medians" of the two distributions are different, not the full distributions. Maybe this is what the authors are looking for (in case they have to explicitly better the null hypothesis), but if they want to test the null hypothesis of equal distributions they should use different nonparametric tests such as the two-sample Kolmogorov Smirnov test, or something similar.

- LINES 302-305. The authors try different classification schemes and they chose the one (decision tree) which is the best performing. Since different classification schemes have different features, I am wondering if the authors have some thoughts on why the decision tree is performing better than the other classification schemes. For instance, is the hierarchical structure of the decision tree which makes the difference? I think that answering to this question may say something important about the precursory activity of the volcano.

* Reference

J. Lindsay, W. Marzocchi, G. Jolly, R. Constantinescu, J. Selva, L. Sandri (2010). Towards real-time eruption forecasting in the Auckland Volcanic Field: application of BET_EF during the New Zealand national disaster exercise 'Rauumoko'. Bull. Volcanol., 72, 185-204.

Reviewer #2 (Remarks to the Author):

This is a well written, topical, and interesting manuscript that will be of great interest to the

community. I feel that the potential implications are greatly overblown in the title and some of the intricacies of the analysis are lost in the abstract to conveniently make it appear more ground-breaking. However, I still believe that the findings are original and valuable, and the paper should be published with some minor modifications, as below:

- The title. This approach has been applied to each of the Whakaari eruptions only, and it is not clear whether it would work for any other volcanic system, monitoring network or eruption style. Therefore, I find the title to be deliberately attention grabbing, and not a true reflection of the method, its application, or findings. Please change.

- The claim within the abstract and elsewhere that 5 of the 7 Whakaari eruptions would have been forecast with this approach. As the model was trained on 6 eruptions in order to provide a forecast for the 7th, this claim is misleading. It implies that the final model would forecast 5 eruptions, whereas it is in fact 5 different models with no logical appreciation of the timeframe being used, i.e. in 'forecasting' a 2013 eruption, data from after 2013 are considered. Therefore, the only true 'forecast' is that of the 2019 eruption, in which past data were used. What would have been interesting is to consider at what point the model became robust enough (i.e. had enough data) to provide forecasts, according to the author's self-determined thresholds. For example, if only data prior to the 2013 eruptions were used, would the eruption consensus be enough to robustly 'forecast' the first 2013 eruption? At what point does it become a method that will forecast future eruptions?

- Three references, for three different eruptions, are assigned to the first line of the main text for more than 1000 people being killed in the last decades. There are two fatality databases available, and reference to those would be more appropriate. The more recent version (Brown et al., 2015) would give the authors values on the numbers of people who died in close proximity and therefore most likely to have been saved with a forecasting approach to blue sky eruptions. [Brown, S.K., Jenkins, S.F., Sparks, R.S.J., Odbert, H. and Auken, M.R., 2017. Volcanic fatalities database: analysis of volcanic threat with distance and victim classification. *Journal of Applied Volcanology*, 6(1), pp.1-20.]

- "Next, we searched for features whose values clearly indicated an association with imminent eruption: these are the precursors." As a reader, I would appreciate a little more detail on how these precursors/features were chosen.

- The missed 2013 and 2016 eruptions. More detail and theories for why these were missed would be valuable. Noted that the 2016 eruption had weaker RSAM features, but what about the 8 August 2013 eruption? And critically, can the authors suggest some physical reasons, i.e. relate the finding back to the volcanic system? Even hypotheses would be valuable rather than providing no information.

- The final paragraph of limitations. A consistent monitoring network is required with no loss of data or change in instrument or location over the time period used to build a training data set. Often, this is the privilege of well-funded volcano observatories and an appreciation of the high level of data and consistency in network required would be an important aspect to note in the final sentences.

- Classification modeling: How many features were eliminated because they were highly correlated? Surely a correlation metric threshold could have been used rather than a visual inspection in order to retain the repeatability of this exercise?

- Figure 4A. The reader needs to see more units between 0 and 100... It looks like the values are 50% or so (i.e. random) until the day or so before the 9 December eruption, except for the model retrained on only eruption 1. Does this imply that at least 2 prior eruptions are required to enable a consensus

for forecast? Or is there something specific about eruption 1? And would this be the same for all eruptions, i.e. returning to second comment. More discussion is required in the supplementary text.

Reviewer #3 (Remarks to the Author):

This manuscript has very many positives, and I feel it offers some excellent insights into how the volcano monitoring community needs to start to analyse and act on its data. The problem, of relatively small, blue-sky eruptions and the impact they can have on people close to the eruption site has been highlighted in recent years, and one approach seems to be to simply shrug the collective volcanologist's shoulders and say 'can't forecast those kind of eruptions'. This manuscript offers possible alternatives, and that is very exciting.

I am not an expert in machine learning, but am familiar with the ideas and some of the procedures commonly used. From that perspective, I am very impressed with the methodology that has been used. The authors seems to have thought of, and tested, all of the questions that came into my mind when reading the methodology, and several that didn't. For their thoroughness they need to be commended.

The limitations of the data set used cannot be avoided. Seismic data provides the only high sample rate data that show any appreciable changes, so are the only possible choice. There are only two seismographs at the volcano, and the authors used data from one of those, arguably the one that provides the most 'noise-free' data. The use of RSAM is an obvious way to get some aggregation of data, yet still offer a lot of opportunity to observe changes over time frames of 10s of minutes to many hours. However, the issue that RSAM is not a direct proxy for tremor intensity is not explained in the manuscript – it is needed; see the comment on the text.

The manuscript correctly states that the model could have provided at least 4 hours warning of the 2019 eruption. But given the false-positive (FP) rate of the model, 1 correct alert out of every 11, would anyone take any notice? I suspect not. I feel the authors have underplayed the consequences of such a high FP rate. Yes, had we used their system for keeping tourists off the volcano it would have only been for 8.5% of the 9 year model window, about one month a year, but after issuing several alerts to stay off the volcano that were not followed by an eruption, I think tour operators and tourists alike would have considered the alerting system too unreliable and ignored it. It could be argued that this study is not the correct place for that kind of discussion, but at least a little effort needs to be put into discussing this – more than stating integration of forecasts into existing systems requires some effort.

The authors refer to the volcano alert level (VAL) on occasions in the text, and extensively in a table in the supplementary material. In mid-2014, the VAL system was changed. It went from one with one level of unrest (VAL 1) to one with two levels of unrest (VAL 1 and 2). This change means referring to VAL before an eruption needs to be done very carefully. The change in the VAL system needs to be noted somewhere in the main text, and in more detail in the table in the supplementary material.

I found the answers to many of my questions about specifics in the supplementary material. I have left those questions as comments in the text where I had them as there may be some opportunity to clarify in the text and not rely solely on the supplementary material. This is the format of the journal and not something the authors have much control over.

There are some additional, specific comments in the text I'd hope the authors would consider.

The provision of the data and analysis codes means that other researchers could reproduce the work.

We are grateful for the thoughtful comments that the reviewers have taken time to provide. We have addressed these in the document below according to the following scheme:

Reviewer comment is italicised

Our direct response is normal font.

XXX: Modified text with line number is in blue. The text changes are also tracked in the revised manuscript document.

Reviewer 1

This paper addresses one important scientific and social issue, such as the forecast of "blue-sky" eruptions; so far, these eruptions are barely predictable causing many serious problems, mainly for tourists.

The paper is very well written and clear; it was a nice reading to me. The technical content of the paper is of high quality and it introduces a novel promising tool to improve our blue-sky eruption forecasting skill.

At the same time, as admitted by the same authors in this manuscript, the evaluation of the forecasting skill of the proposed method is still very speculative, being based on hindcasting, which often leads to significant overfitting of the data and too optimistic conclusions.

Addressing the issue of overfitting is something we have done very carefully. To illustrate that problem and how we deal with it, we have relaxed the aspects of machine learning workflows that guard against overfitting. The result is a simplified trigger model with two features (a two node decision tree) and trigger thresholds optimised on the entire data set. Its performance on the previous data is superior to the ensemble model and it provides a discussion point on overfitting, robustness and future model testing.

201: In automating key decisions such as feature (physics) selection, classifier calibration, and cross-validation, we guard against model overfitting. To illustrate the costs of these choices, we developed a competing forecast model in which we hand-picked two features (RSAM maximum and 4-hr harmonic, Fig. 1D-E) and issued an alert whenever they both exceeded thresholds. This two-node decision tree is optimized on the entire dataset, i.e., with no cross-validation or pseudoprospective testing (supp. Fig. 5). This manual model issued twelve alerts in the 9-yr study period, which contained six out of the seven eruptions (Apr 2016 is missed). The alert duration was 55 days and the eruption probability during alert was 50%. Superficially, this performance is significantly better than the ensemble model. However, as overfitting has been maximized, its prospective performance on future data is likely to be very poor.

So, in general, it is important to make this distinction crystal clear for the readers, also in the abstract; to me this would not downplay the possible impact of the paper. As a matter of fact, the paper is more methodological and the testing phase is still very preliminary. I understand that it takes time to evaluate a model in a prospective way (seismologists have exactly the same problem with large earthquakes), but it is the gold rule to evaluate the real predictive skill, and I do not know any shortcut.

We have clarified the nature of the pseudoprospective testing so that the claimed accuracy can be understood in that context. In addition, we have made clear that the next step is formal prospective testing on future data.

17: We developed a model to recognize these precursors and issue an alert indicating an elevated likelihood of imminent eruption. Cross-validating this model on five eruptive episodes, the model was able to issue an advanced alert for an unseen eruptive episode in four instances. For the fatal 2019 eruption, the model would have issued at least four hours warning. This performance and its

robustness on previous data makes a strong case to adopt prospective testing of real-time forecasting models on future data at other similar volcanoes.

Below I list the specific comments on this paper

- The title probably is too emphatic, in the sense that the paper addresses one specific kind of eruptions (blue-sky) which are very interesting from the scientific point of view, but usually impact a limited number of tourists. It goes without saying that this is not meant to under-evaluate the life of each single person, but probably the word "catastrophe" may be used for different types of eruptions.

We have changed the title to:

2: “Automatic precursor recognition and real-time forecasting of sudden explosive volcanic eruptions”

*- As corrected mentioned by the authors, the VALS in many volcanoes (and certainly for Whakaari volcano) are *not* a forecasting scheme, even though it is often interpreted as such. Actually, in the volcanological literature it is possible to find critics to VALS (and more in general on how volcanologists manage phases of unrest), because under a VALS approach volcanologists tend to act as decision makers without having proper competences to decide, for example, "when" it is the right moment to call for an evacuation and how long it should last; I think we, as scientists/volcanologists, have to pay attention to these aspects if we do not want to face harsh critics and possible legal prosecutions after any kind of subsequent formal enquiry (it would not be the first time in natural hazard...). So I think we, as scientists/volcanologists, should be very careful in suggesting any kind of "consensus" thresholds in the forecasts (see LINES 177-178), because they do not have any scientific meaning but they can be justified only in a decision-making world, e.g. through some sort of cost/benefit analysis for risk mitigation decisions. Of course, the method proposed by the authors can be integrated in a VAL, but I would make very clear the distinction between the forecasting tool proposed by the authors and the integration into a VALS. This mapping requires different competences. The authors seem to recognize this crucial point at lines 139-140, but at the end of the paper (LINES 171-180), it seems that they advocate some thresholds chosen through a consensus (of volcanologists? who is the "designer" that has to chose this?).*

To avoid confusion with the more common “consensus amongst a group of decision makers”, we have replaced the term with ensemble mean.

112: the ensemble mean prediction is used to issue alerts.

However, this is a good opportunity to address the challenges of the “who decides?” problem.

226: This forecasting approach does not remove the question of “who decides” about warning. The forecast model issues a warning with an estimated eruption probability that is conditional upon a specified threshold. Choosing this value is a tradeoff between alert duration, accuracy, and specificity, with implications for public safety, economic activities and public trust. Deciding who is responsible for setting these thresholds is a societal debate, as is determining access to the model outputs.

Ultimately, tour operators, government regulators and the public all bear some responsibility for adjusting their actions in response to new information about the volcano state.

- LINES 104-107. The authors acknowledge the problem of handling sequences in which we have many more "no eruption" than "eruption" windows. There is not a unique way to handle this problem (for instance, one possible strategy is to weigh in a different way the correct predictions and the false alarm rates for each category). The authors decide to take 100 subsets in which the "no eruptions" windows are randomly discarded to make the numerosity of each class more comparable. It is not clear to me how this step is carried out. What is the proportion of "no eruption" windows that

are discarded for each subset?

If we want to have similar number of "eruption" and "no eruption" windows I guess that the percentage of discarded windows is very large (even though the authors say they discard "some no eruption windows"). In any case I am wondering if different choices of the percentage of discarded windows may affect the results.

In the model presented, eruption/no-eruption windows are a 16/21 split (43% : 57%) in each subset (misreported as 16/24 or 40% : 60% split). With 6437 non-eruptive windows in each training set, 21 are randomly selected without replacement and the other 6416 are discarded. We have added a new supplementary figure to compare the forecast model for different values of this split. The principal result is that increasing non-eruptive windows reduces specialisation on recognising eruptions, but also non-eruptive periods of volcanic unrest.

342: 6453 entries in Y_{train} of which only 16 were labeled 1. We rebalanced the dataset by randomly sampling (without replacement) 21 of the label 0 windows (0.3%) from M_{train} and Y_{train} and discarding the rest. Features and labels of the reduced set of $N_{ru} = 37$ windows (supp. Fig. 2C) are denoted $[M_{ru}, Y_{ru}]$. The undersampling ratio is the ratio of eruptive windows to non-eruptive windows, in this case 0.75.

447: The performance of the model forecast is similar under variation of the undersampling ratio (supp. Fig. 9). Reducing this value includes more non-eruptive data in the training subsets, which decreases model alert duration. This is similar to the trend identified in supp. Fig. 8B, i.e., a reduced specialization on eruptive data means that fewer alerts are identified during non-eruptive episodes of volcanic unrest.

new supplementary figure 9

522: Fig 9: A. Ensemble mean from a forecast model in the period preceding the Dec 2019 eruption, for different values of the undersampling ratio. B. Performance of the different forecast models, in terms of confidence of imminent eruption (crosses) and total alert duration (circles).

- LINES 145-149. The authors say that data-driven models can only anticipate eruptions whose precursors have been seen before. The authors forget that, nowadays, many eruption forecasting schemes are not full data-driven models. For example, many implementations of the Bayesian Event Tree model take already into account the possibility to have different precursors from eruption to eruption, and precursors that have not been observed before.

We have added a note to address this issue.

175: Adding precursory signals from other volcanic systems (e.g., via Bayesian approaches³³) could also help, although this would require testing and evaluation to ensure the unique signals identified for a particular volcanic system are not diluted.

- LINES 163-170. The authors emphasize the importance to have "real-time" tools to forecast the evolution of the volcanic system, compared to the slow process of getting forecasts from experts' judgment elicitation. I do agree with that. But probably it should be said that this point has been raised several times in the past, also for New Zealand volcanoes (Lindsay et al., 2010).

We have integrated this reference in the text

196: The result could be promptly uploaded to a public website or, as appropriate, communicated directly to VAL operators, which is valuable both prior to and during a volcanic crisis³⁷.

- LINES 280-283. The authors use the Mann-Whitney U test to check if two distributions are different.

I think that the Mann-Whitney U test is tailored to check if the "medians" of the two distributions are different, not the full distributions. Maybe this is what the authors are looking for (in case they have to explicit better the null hypothesis), but if they want to test the null hypothesis of equal distributions they should use different nonparametric tests such as the two-sample Kolmogorov Smirnov test, or something similar.

Thanks for pointing this out. We have revised the interpretation of rejected null-hypotheses for the Mann-Whitney U test. However, both tests, Mann-Whitney U and Kolmogorov Smirnov have been designed for testing that two distributions are different. Overall, we have found that the Mann-Whitney U test is more sensitive to detect differences between distributions with similar shape but shifted location, which – of course – corresponds to your remark.

334: were likely to have been drawn from a distribution having a different median than the distribution labeled 0

- LINES 302-305. The authors try different classification schemes and they chose the one (decision tree) which is the best performing. Since different classification schemes have different features, I am wondering if the authors have some thoughts on why the decision tree is performing better than the other classification schemes. For instance, is the hierarchical structure of the decision tree which makes the difference? I think that answering to this question may say something important about the precursory activity of the volcano.

We think you are suggesting that branching decision trees perform well when there are multiple signals to be classified, in much the same way as event trees are favoured in other eruption forecasting studies. We agree, although demonstrating the truth of that claim can be tricky. However, because we are training an ensemble of decision trees, which classify a balanced problem containing the same true positives but different true negatives, we basically implement a variant of a random forest classifier. We have added this point to our manuscript and added a note about the alignment with event tree methods.

111: This bagging approach implements a random forest specialized on eruptive windows,

360: Possibly this is because, like event trees³², the hierarchical structure of decision trees can naturally accommodate different kinds of eruption precursors.

* Reference

J. Lindsay, W. Marzocchi, G. Jolly, R. Constantinescu, J. Selva, L. Sandri (2010). Towards real-time eruption forecasting in the Auckland Volcanic Field: application of BET_EF during the New Zealand national disaster exercise 'Ruaumoko'. Bull. Volcanol., 72, 185-204.

Reviewer 2

This is a well written, topical, and interesting manuscript that will be of great interest to the community. I feel that the potential implications are greatly overblown in the title and some of the intricacies of the analysis are lost in the abstract to conveniently make it appear more groundbreaking. However, I still believe that the findings are original and valuable, and the paper should be published with some minor modifications, as below:

- The title. This approach has been applied to each of the Whakaari eruptions only, and it is not clear whether it would work for any other volcanic system, monitoring network or eruption style. Therefore,

I find the title to be deliberately attention grabbing, and not a true reflection of the method, its application, or findings. Please change.

We have changed the title to:

2: “Automatic precursor recognition and real-time forecasting of sudden explosive volcanic eruptions”

- The claim within the abstract and elsewhere that 5 of the 7 Whakaari eruptions would have been forecast with this approach. As the model was trained on 6 eruptions in order to provide a forecast for the 7th, this claim is misleading. It implies that the final model would forecast 5 eruptions, whereas it is in fact 5 different models with no logical appreciation of the timeframe being used, i.e. in ‘forecasting’ a 2013 eruption, data from after 2013 are considered. Therefore, the only true ‘forecast’ is that of the 2019 eruption, in which past data were used.

This claim is quite clearly qualified as an outcome of pseudoprospective testing but we accept that the terminology will not be familiar to many readers and so have replaced it with an explanation

17: We developed a model to recognize these precursors and issue an alert indicating an elevated likelihood of imminent eruption. Cross-validating this model on five eruptive episodes, the model was able to issue an advanced alert for an unseen eruptive episode in four instances. For the fatal 2019 eruption, the model would have issued at least four hours warning.

What would have been interesting is to consider at what point the model became robust enough (i.e. had enough data) to provide forecasts, according to the author’s self-determined thresholds. For example, if only data prior to the 2013 eruptions were used, would the eruption consensus be enough to robustly ‘forecast’ the first 2013 eruption? At what point does it become a method that will forecast future eruptions?

The question of robustness is a good one and is a topic we should have addressed more explicitly. There are two aspects to be demonstrated: (1) robustness of our selected model in forecasting of the data, and (2) robustness compared to other models.

On the first topic, per your suggestion, we have included a matrix in supplementary material that essentially runs these calculations: training and freezing a forecaster on the first N eruptions, and then predicting the remaining at a consensus threshold of 80%. The results indicate that the Oct 2013 and Dec 2019 eruptions were forecastable in practice, whereas the Aug 2013 and 2016 events were not. So the model appears to be somewhat robust from an early stage and with not many eruptions to learn from. We have also distinguished between historical performance of forecasting in practice vs the pseudoprospective test that offers an estimate of the present performance.

154: The “leave one out” method of performance evaluation only yields a theoretical estimate of the model’s present accuracy based on all data. However, if the forecast model were in operation since the 2012 eruption, its performance would be different because it would have had access to less training data. To emulate this scenario, we trained four models using data up to one month prior to each of the eruptive periods two (Aug 2013) through five (Dec 2019), and then generated a forecast for the eruptive period (supp. Fig. 4). Both the Oct 2013 and Dec 2019 eruptions are anticipated at a 0.8 alert threshold based on knowledge of previous events. In contrast, observation of the 2012 eruption is not sufficient to pick the Aug 2013 eruption.

new supplementary figure 4

489: Fig. 4: Matrix of retrospective forecasting outcomes. Rows indicate forecast models trained using consecutive eruptive periods, e.g., row two is trained using the 2012 and Aug 2013 eruptions and data up to one month before the Oct 2013 eruptive period. Columns indicate the performance of a forecaster on a future eruptive period as measured by model confidence, which is the maximum

ensemble mean in the 48 hours prior to the eruptive period. For example, row three, column five indicates that a forecast model trained using the 2012 and two 2013 eruptive periods will anticipate the 2019 eruption with a confidence of 0.91. On the second point of robustness compared to alternative models, we would point to the aspects of machine learning pipelines that promote predictive performance: automatic feature selection, automatic model calibration, and cross-validation/pseudoprospective testing. To emphasise those elements, we have developed a second model that drops all three in favour of our own decision making. We chose the features and the triggering thresholds using full knowledge of the eruption data. As might be expected, that model's performance on the past data is significantly better. However, we expect it would be fragile when generalising to future data.

201: In automating key decisions such as feature (physics) selection, classifier calibration, and cross-validation, we guard against model overfitting. To illustrate the costs of these choices, we developed a competing forecast model in which we hand-picked two features (RSAM maximum and 4-hr harmonic, Fig. 1D-E) and issued an alert whenever they both exceeded thresholds. This two-node decision tree is optimized on the entire dataset, i.e., with no cross-validation or pseudoprospective testing (supp. Fig. 5). This manual model issued twelve alerts in the 9-yr study period, which contained six out of the seven eruptions (Apr 2016 is missed). The alert duration was 55 days and the eruption probability during alert was 50%. Superficially, this performance is significantly better than the ensemble model. However, as overfitting has been maximized, its prospective performance on future data is likely to be very poor.

new supplementary figure 5

498: Fig. 5: Alternative parameter set for the forecast model illustrating potential overfitting. Restricted to two user-selected features – RSAM maximum (magenta) and 4 hr harmonic (blue) – and a two node decision tree with user-selected thresholds that were optimised using knowledge of the entire dataset. The forecast model generates twelve alert periods that cover six eruptions, missing only the Apr 2016 event, and the total in alert duration is 1.7% of the analysis period (56 days).

- *Three references, for three different eruptions, are assigned to the first line of the main text for more than 1000 people being killed in the last decades. There are two fatality databases available, and reference to those would be more appropriate. The more recent version (Brown et al., 2015) would give the authors values on the numbers of people who died in close proximity and therefore most likely to have been saved with a forecasting approach to blue sky eruptions. [Brown, S.K., Jenkins, S.F., Sparks, R.S.J., Odbert, H. and Auken, M.R., 2017. Volcanic fatalities database: analysis of volcanic threat with distance and victim classification. Journal of Applied Volcanology, 6(1), pp.1-20.]*

This is a good reference to use and we have included it at the suggested location.

- *“Next, we searched for features whose values clearly indicated an association with imminent eruption: these are the precursors.” As a reader, I would appreciate a little more detail on how these precursors/features were chosen.*

We have adjusted the wording in the following sentences to clarify how the selection occurs (with the mathematical details being reserved for the methods section).

95: This was done by applying a statistical test to the distribution of each feature (over all windows) and identifying those features whose values tagged with a 1 were significantly different to those tagged 0 (see methods for details).

- *The missed 2013 and 2016 eruptions. More detail and theories for why these were missed would be valuable. Noted that the 2016 eruption had weaker RSAM features, but what about the 8 August 2013*

eruption? And critically, can the authors suggest some physical reasons, i.e. relate the finding back to the volcanic system? Even hypotheses would be valuable rather than providing no information.

We have added some discussion for why these two events were missed and expanded the implications of missing them.

144: The middle eruption in this sequence had anomalously high RSAM activity in the 24 hours prior (supp. Fig. 3A) that would likely have concerned a human operator. However, the signature differed from other eruption precursors enough that the forecast model did not issue an alert.

163: The missed 2016 eruption is much weaker at 40-min and 4-hr RSAM periodicity that characterizes the other eruptions (Fig. 1D-F), and exhibited low tremor before and after the event (supp. Fig. 3B). The VAL prior to this eruption noted minor volcanic unrest and gas flux and surface temperatures were normal or slightly elevated, in contrast to the 2012 and 13 eruptions levels of gas flux and surface activity were rising notably (supp. Tab. 1). A recent study³² of ejecta from the 2016 eruption found evidence of hydrothermal mineralization and a hydraulic seal in the shallow vent area, along with no magmatic ejecta. These factors indicate that the event was a pure hydrothermal explosion of trapped meteoric fluids, without any increased injection of fluids into the system from below.

³²Kennedy, B. M. et al. Pressure Controlled Permeability in a Conduit Filled with Fractured Hydrothermal Breccia Reconstructed from Ballistics from Whakaari (White Island), New Zealand. *Geosciences* 10, 138 (2020).

new supplementary figure 3

484: Fig. 3: RSAM (black), MF (blue) and HF (green) data streams for two missed eruptions. The timing of eruption are indicated by the vertical red bars. A. The 8 Oct 2013 eruption, the middle of three during the eruptive period. B. April 2016 eruption.

- The final paragraph of limitations. A consistent monitoring network is required with no loss of data or change in instrument or location over the time period used to build a training data set. Often, this is the privilege of well-funded volcano observatories and an appreciation of the high level of data and consistency in network required would be an important aspect to note in the final sentences.

All good points and we have mentioned them. A counterpoint is that this study relies on a single seismometer rather than an extensive network and this is potentially a strength for applying the system in developing countries.

216: This is on top of the resourcing necessary to install and maintain a telemetered seismic station at a remote location for extended periods. Extensive monitoring networks are the privilege of well-funded volcano observatories, but this system can run only with a single station to reduce costs.

- Classification modeling: How many features were eliminated because they were highly correlated? Surely a correlation metric threshold could have been used rather than a visual inspection in order to retain the repeatability of this exercise?

The dropped features are hard-coded in the software accompanying this submission so the approach is repeatable. These dropped features were different ways of computing linear trends. 67 out of 700 features were dropped this way.

364: Highly correlated features were due to different methods for calculating linear trends and we eliminated these by dropping their columns (53 features) from *M* and retraining the classifiers.

- Figure 4A. The reader needs to see more units between 0 and 100... It looks like the values are 50% or so (i.e. random) until the day or so before the 9 December eruption, except for the model retrained on only eruption 1. Does this imply that at least 2 prior eruptions are required to enable a consensus

for forecast? Or is there something specific about eruption 1? And would this be the same for all eruptions, i.e. returning to second comment. More discussion is required in the supplementary text.

The figures have been adjusted as suggested. There is a misunderstanding here, because we have not been clear enough that model consensus is not interpretable as a probability. That would require a careful calibration exercise, which itself may not yield a reliable probability. Instead, we have preferred a simplistic but more comprehensible expression of probability as the proportion of all alerts that contain eruptions.

380: We cannot interpret this as a probability of a future eruption, because the models are not individually or collectively calibrated.

402: We then defined the eruption probability during alert as $TP/(FP+TP)$, which is the probability that an eruption will occur within a randomly selected alert.

In terms of forecasting performance in a more normal sequence of eruption training, we refer to our response to your earlier comment (and the new figures and discussion included there).

Reviewer 3

This manuscript has very many positives, and I feel it offers some excellent insights into how the volcano monitoring community needs to start to analyse and act on its data. The problem, of relatively small, blue-sky eruptions and the impact they can have on people close to the eruption site has been highlighted in recent years, and one approach seems to be to simply shrug the collective volcanologist's shoulders and say 'can't forecast those kind of eruptions'. This manuscript offers possible alternatives, and that is very exciting.

I am not an expert in machine learning, but am familiar with the ideas and some of the procedures commonly used. From that perspective, I am very impressed with the methodology that has been used. The authors seems to have thought of, and tested, all of the questions that came into my mind when reading the methodology, and several that didn't. For their thoroughness they need to be commended.

The limitations of the data set used cannot be avoided. Seismic data provides the only high sample rate data that show any appreciable changes, so are the only possible choice. There are only two seismographs at the volcano, and the authors used data from one of those, arguably the one that provides the most 'noise-free' data. The use of RSAM is an obvious way to get some aggregation of data, yet still offer a lot of opportunity to observe changes over time frames of 10s of minutes to many hours. However, the issue that RSAM is not a direct proxy for tremor intensity is not explained in the manuscript – it is needed; see the comment on the text.

The manuscript correctly states that the model could have provided at least 4 hours warning of the 2019 eruption. But given the false-positive (FP) rate of the model, 1 correct alert out of every 11, would anyone take any notice? I suspect not. I feel the authors have underplayed the consequences of such a high FP rate. Yes, had we used their system for keeping tourists off the volcano it would have only been for 8.5% of the 9 year model window, about one month a year, but after issuing several alerts to stay off the volcano that were not followed by an eruption, I think tour operators and tourists alike would have considered the alerting system too unreliable and ignored it. It could be argued that this study is not the correct place for that kind of discussion, but at least a little effort needs to be put into discussing this – more than stating integration of forecasts into existing systems requires some effort.

We are pleased for the opportunity to engage this conversation. This is a communication issue. It is perhaps better that we avoid “alarm” and “false alarm” terms, because our alert periods do not

guarantee eruption. Rather, by estimating an eruption probability during alert, we can be fairly assessed over long term performance. The general public understand percentages well enough that a 10% likelihood of eruption is probably large enough to stop people taking risks with their lives, but not so large that it feels like an assurance has been given.

123: Not all alerts will contain eruptions, so the true positive rate is an estimate for the probability that an eruption will occur during an alert. This is an important figure for public communication.

148: At a 0.8 threshold, the model issued 58 alerts that covered 8.5% of the 9-year study period. With five of the alerts containing eruptions, the eruption probability during an alert is estimated at 8.6%. A more specific model, using a 0.93 threshold, generates 26 alert periods covering 2.5% of the study period, with a corresponding 15% eruption probability during alert.

The authors refer to the volcano alert level (VAL) on occasions in the text, and extensively in a table in the supplementary material. In mid-2014, the VAL system was changed. It went from one with one level of unrest (VAL 1) to one with two levels of unrest (VAL 1 and 2). This change means referring to VAL before an eruption needs to be done very carefully. The change in the VAL system needs to be noted somewhere in the main text, and in more detail in the table in the supplementary material.

Good point. The change in VAL is now described in the supplementary Table header.

459: Note, prior to July 2014, New Zealand operated a VAL system with one unrest level (VAL 1) and four categories of eruption (VAL 2-5). After July 2014, this was replaced with by a scheme with two unrest levels (VAL 1-2) and three eruptive levels (VAL 3-5)³.

I found the answers to many of my questions about specifics in the supplementary material. I have left those questions as comments in the text where I had them as there may be some opportunity to clarify in the text and not rely solely on the supplementary material. This is the format of the journal and not something the authors have much control over.

Thank you. We have addressed these comments below:

It is not stated what 8 to 14% represents. It is median to 84th percentile (with best estimate counted twice and upper and lower bounds counted once each). If these numbers are to be included, then what they represent needs to be somewhere too.

We have captured this and provided a reference.

31: Expert elicitation^{10,11} was used to estimate an eruption likelihood of 8 to 14% for the 4-week period beginning 2 Dec 2019, one week prior to the eruption¹², which represents the 50th to 84th percentile range of estimates of the expert panel.

¹¹Deligne, N. I., Jolly, G. E., Taig, T. & Webb, T. H. Evaluating life-safety risk for fieldwork on active volcanoes: the volcano life risk estimator (VoLREst), a volcano observatory's decision-support tool. *Journal of Applied Volcanology* 7, 7 (2018).

Tremor would normally be described as 'volcano' rather than 'volcano-tectonic'.

Fixed.

Need to state the time interval for the calculated values. Value every xxx secs/mins/hrs.

RSAM, and other measures are not direct measures of volcanic tremor. The 2-5 Hz band is where tremor at White Island most commonly has most of its energy. But RSAM in this band is not a direct

proxy for the tremor intensity. Others signals do have energy in this band and this will cause non-tremor noise in RSAM.

We have changed the wording to clarify that the data are capturing parts of the tremor signal.

74: We processed nine years of data, from 1 Jan 2011 to 1 Jan 2020 (Fig. 1B), to obtain four time series that capture different parts of the tremor signal (RSAM, DSAR, medium and high frequency bands – MF and HF – sampled every 10 minutes; see methods).

Meaning not clear. There were 769 features that applied to a single 48 hr window, or features were calculated for parts of the 48 hr window?

We have added a clarification

88: First, we sliced the tremor data into 48-hour windows that overlapped by 36 hours. Second, for each window, we computed 700 features of the 48-hr time series

Would be interested so see what range of frequencies were calculated.

Sampling rate is needed for this, so we are sure there is no aliasing.

I have added these details into the methods section. FFTs are calculated for each window, so the range of frequencies is $1/T$ to $n/2T$, where T is the window length, and n is the number of samples (288 for 10 min samples in 48 hr windows). So freq. range is 8.3×10^{-4} to 5.8×10^{-6} Hz.

327: For a window with n samples, FFT coefficients are computed for frequencies $[1, \dots, n/2 - 1] / (T_w n)$, which for a 48-hr window with 10 minute samples corresponds to 8.3×10^{-4} to 5.8×10^{-6} Hz.

Normally data are divided into a training set and a testing set; very difficult with only 7 eruptions. This is not mentioned at all – what was done in this case?

As you mention in a later comment, this is described later. We have also now referred to train-test split in the abstract.

19: Cross-validating this model on five eruptive episodes, the model was able to issue an advanced alert for an unseen eruptive episode in four instances.

~1600 windows (for zero overlap, as overlap not stated), and 7 eruptions. What proportion were discarded?

We have now stated the overlap, and there are about 6400 windows. This means, to train one classifier, we keep only 0.3% of the non-eruptive windows. But this step is repeated 100 times, so across all classifiers, about 30% of the non-eruptive windows are used. We have tested using 1000 classifiers, in which case nearly all non-eruptive windows are used, but the results are not very different when using 100 models. These details now appear in the methods.

347: Across all the classifiers, about 30% of the non-eruptive data is used.

This answers my previous question, values calculated every 10 mins. Does this also means that the overlap between 48 hr windows is 47 hr 50 min? A 40 hr rolling window advancing 10 mins each step?

This comment and others related to window overlap, sampling rate, and forecast update period have been addressed throughout the manuscript.

89: we sliced the tremor data into 48-hour windows that overlapped by 36 hours

75: that capture different parts of the tremor signal (RSAM, DSAR, medium and high frequency bands – MF and HF – sampled every 10 minutes; see methods)

387: This is important because if a precursor occurs less than 12 hours before an eruption, it could be missed by a forecast model updating every 12 hours.

By this you mean that you removed the data 1 month either side of the eruption and retrained/ tested the model? Or was that done only once, with the repeated part being calculation of threshold and hence false alarm rate?

Certainly the former, we have added a note to clarify.

134: We then repeated these steps for the other eruptive periods in the dataset, which yielded five separately trained models and forecasting tests (Fig. 2C).

As a 'VAL operator' I would struggle to use alerts in my evaluation of the likelihood on an eruption if I knew the false positive rate was 10 out of 11 times.

Many of the reviewer's comments (quite reasonably) address the issue of a high false positive rate. The issue was raised by other reviewers as well, and our response is that this is primarily a communication issue. Therefore, we have revised the manuscript to avoid "false alert" terminology, acknowledging that our accuracy is not high enough to offer an "assurance of eruption".

Are the authors suggesting that it would be okay to announce that an eruption was likely, event though the false-positive rate is so high, as the possible eruption window is only 8.5% of the time?

We do not suggest an eruption is likely, although we accept that using words like "alert" and "alarm" might convey this to the general public. Therefore, we have instead computed and use primarily a "probability of eruption during alert".

123: Not all alerts will contain eruptions, so the true positive rate is an estimate for the probability that an eruption will occur during an alert. This is an important figure for public communication.

148: At a 0.8 threshold, the model issued 58 alerts that covered 8.5% of the 9-year study period. With five of the alerts containing eruptions, the eruption probability during an alert is estimated at 8.6%.

I'm interested, in this context, what the authors might consider as 'perfect data'. This means data that will always provide an eruption forecast and a nil false positive rate?

The data and model we present together are unable to capture all eruptions with no false positive. This is the goal of any forecaster. Inaccuracy is probably due to shortcomings in *both* the data and the models.

173: Future refinements to address this issue could include training the model using other data streams, e.g., continuous gas monitoring or shallow sensors installed near the vent.

While these comments are quite valid, they are sort of irrelevant as the study uses on seismic data.

In our opinion, discussion of the broader data around missed eruptions is warranted here.

A revised VAL system was introduced on 2014-07-01. Five eruptions occurred before this time, and two after.

2012/13 eruptions contrasted occurred when only one unrest VAL: 1. Discussion of VAL in this context is dangerous as readers are unlikely to know VAL system was revised, and when.

The revision of the VAL system mentioned by the reviewer does not seem particularly consequent given that (1) it was set to VAL 1 prior to all eruptions except the most recent and (2) VAL 1

corresponded to “Signs of volcano unrest” prior to Jun 2014 and “Minor volcanic unrest” afterward. As the reviewer has suggested, we have noted that this revision occurred in Table 1 summarising VAL changes, and the readers are welcome to explore the cited references for more information.

459: Note, prior to July 2014, New Zealand operated a VAL system with one unrest level (VAL 1) and four categories of eruption (VAL 2-5). After July 2014, this was replaced with by a scheme with two unrest levels (VAL 1-2) and three eruptive levels (VAL 3-5)3.

Which is exactly what ‘expert judgement to set alert levels’ does, isn’t it?

Yes. As we have said throughout, we propose that our scheme sit alongside and support VAL, not supplant it.

I don’t think that VAL operators are influenced by either social or political pressures, certainly not in NZ.

This is not intended as a specific comment about the New Zealand system. However, in our view, all humans making decisions that affect the safety and economic prosperity of others are likely to feel these pressures – real or imagined – to some degree.

This analysis suggests that there is a time-invariant probability of an eruption whenever there is an automatic alert. As noted earlier, the VAL operators also provide, by means of expert elicitation, an estimate of eruption probability, and this varies quite substantially from time to time, which is more realistic than a time-invariant probability

Comparison against the expert elicitation method for the generation of probabilities is not a central focus of this paper. A post-hoc analysis of the expert probabilities issued by VAL operators following the 2019 eruption paints an unflattering picture of the elicitation methodology. So while it may appear more realistic to have a probability that changes with time, that should not be confused as an indicator of quality or accuracy.

At White Island there are few options for data to chose, so designer bias in data selection is not that important.

Agreed, although this was intended as a general comment on possible sources of biases with these kinds of approach.

Is the time shift between window i and window $i+1$ correct as shown, or is it just indicative?

This is accurately depicted.

This illustrates an observation that the authors maybe should include on the text somewhere – the ‘background’ % eruption consensus. In the period illustrated, this is over 50% for much of the time. Does this not suggest that the % eruption consensus is not a strong ‘predictor’?

We have clarified in the text that the output of the ensemble model is itself not a predictor, because it has not been calibrated to yield probabilities. It is used as an input to the decision model, and it is these alerts (and the probabilities calculated from them) that should be interpreted.

380: We cannot interpret this as a probability of a future eruption, because the models are not individually or collectively calibrated.

402: We then defined the eruption probability during alert as $TP/(FP+TP)$, which is the probability that an eruption will occur within a randomly selected alert.

If the alert threshold is too high the 2019 eruption could have been missed and we’d still get 7/10 false-positives. I cannot see how such a high FP rate could be workable in any practical situation.

Again, I think this comes back to the messaging. If we tell people that an eruption “absolutely will occur shortly” and we are wrong 90% of the time, they will rightly doubt us. If we tell them, “there is a 10% likelihood of an eruption in the next few days” and then no eruption occurs, I think the public will take a reasonable view of that.

Comparing B with Figure 1, it looks like periods with many FPs occurred when tremor was strong, and/or when it increased from a prolonged low period. My skeptical side says just call an alert when tremor is strong!

We can produce such a forecast model by intervening at various steps in our workflow to make choices otherwise devolved to algorithms. The result is a simple triggering model that indeed has better performance statistics but is also likely to be overfit and non-robust.

201: In automating key decisions such as feature (physics) selection, classifier calibration, and cross-validation, we guard against model overfitting. To illustrate the costs of these choices, we developed a competing forecast model in which we hand-picked two features (RSAM maximum and 4-hr harmonic, Fig. 1D-E) and issued an alert whenever they both exceeded thresholds. This two-node decision tree is optimized on the entire dataset, i.e., with no cross-validation or pseudoprospective testing (supp. Fig. 5). This manual model issued twelve alerts in the 9-yr study period, which contained six out of the seven eruptions (Apr 2016 is missed). The alert duration was 55 days and the eruption probability during alert was 50%. Superficially, this performance is significantly better than the ensemble model. However, as overfitting has been maximized, its prospective performance on future data is likely to be very poor.

new supplementary figure 5

498: Fig. 5: Alternative parameter set for the forecast model illustrating potential overfitting. Restricted to two user-selected features – RSAM maximum (magenta) and 4 hr harmonic (blue) – and a two node decision tree with user-selected thresholds that were optimised using knowledge of the entire dataset. The forecast model generates twelve alert periods that cover six eruptions, missing only the Apr 2016 event, and the total in alert duration is 1.7% of the analysis period (56 days).

Is such a high discard rate normal, considered acceptable? What are the effects of this? Does ensemble analysis deal with any detrimental effects of this discarding?

We have added a supplementary figure testing the effect of the undersampling ratio. It is necessary if the classifier is not to be overwhelmed by non-eruptive data. Validation of these kinds of decisions comes through pseudoprospective testing, in which our model performs very well.

447: The performance of the model forecast is similar under variation of the undersampling ratio (supp. Fig. 9). Reducing this value includes more non-eruptive data in the training subsets, which decreases model alert duration. This is similar to the trend identified in supp. Fig. 8B, i.e., a reduced specialization on eruptive data means that fewer alerts are identified during non-eruptive episodes of volcanic unrest.

This is really good.

Thank you.

Given that the decision tree algorithm still has such a large FP rate, the reader might want an idea of the FP rate for other algorithms.

Alert duration, plotted on the righthand axis of Fig 7B, provides a proxy for this rate.

512: Total alert duration increases with the number of issued alerts and therefore is a proxy for the false positive rate.

I do not claim to understand the all the implications of the change in window size and forecast frequency, but I wonder if this has any unexpected consequences. Can the authors assure the readers that it is valid to change the window size and forecast frequency at this stage of the analysis.

The window size is not changed as the model always receives 48-hr of data. Therefore, the ensemble model remains the same, but it is more frequently called upon to evaluate data. The principal effect is that alerts are issued more frequently, because there are more opportunities for $\bar{y} > \bar{y}_{crit}$. This is important – if there is only an 8-hr gap between a precursor and an eruption, a forecast model re-evaluated every 12-hrs would potentially miss this.

387: This is important because if a precursor occurs less than 12 hours before an eruption, it could be missed by a forecast model updating every 12 hours.

This doesn't see that high, to me. Are here other examples, in similar fields, of higher MCC scores?

We have not come across any useful comparisons here, however, we have added additional description to this metric.

396: The MCC for our model varies between 0.2 and 0.4 (Fig. 2A) depending on \bar{y}_{crit} , indicating a weak positive to moderate positive relationship.

Can this comment be explained more? Is it suggesting that the public would expect false alerts and missed eruptions to be more critically important?

Our suggestion is that the general public are likely to tolerate (assign lower weight to) false alerts than to a missed eruptions, whereas the MCC treats those two equally. We have clarified this

396: As a metric, MCC assigns the same weight to a missed eruption as it does to an alert that does not contain an eruption. In practice, we expect that the general public are likely to assign greater significance to missed eruptions.

I think this explains my earlier question. Sadly most readers won't go through all the supplemental material. This should be clarified in the main text, I think.

We have added some more text to clarify

129: The test data were then input to the forecaster that returned an ensemble mean prediction over the two-month period. Only prediction of the eruption in the test data is meaningful.

And probably a 'workable' interval for anyone issuing a public alert.

Agreed, although perhaps not our place to suggest this.

I presume this refers to the DT model, though it doesn't say.

We have included a comment in the methods

358: The decision tree consistently outperformed other classifiers in model confidence of imminent eruption while also minimizing total alert duration (supp. Fig. 7). Possibly this is because, like event trees³³, the hierarchical structure of decision trees can naturally accommodate different kinds of eruption precursors. Only results using decision tree classifiers are reported here.

The provision of the data and analysis codes means that other researchers could reproduce the work.

REVIEWERS' COMMENTS:

Reviewer #1 (Remarks to the Author):

The authors addressed satisfactorily all the comments of my review.

Reviewer #2 (Remarks to the Author):

I appreciated the thoughtful response to reviewers and the changes implemented for all reviewer comments. I believe that this has made the manuscript much stronger and clearer for a broad audience. I have no further suggestions and believe the manuscript is now ready for publication.

Reviewer #3 (Remarks to the Author):

I have reviewed the authors responses to my comments and am satisfied that they have been satisfactorily addressed. The manuscript is much improved. The authors also seem to have responded to the comments of the others authors.